# Prevalence of short interpregnancy interval and its associated factors among pregnant women in Debre Berhan town, Ethiopia

**Hana Mamo[1], Abinet Dagnaw[2], Nigussie Tadesse Sharew[2], Kalayu Brhane[2,3], Kehabtimer Shiferaw Kotiso[4] ***

**1** Department of Public Health, College of Medicine and Health Sciences, Wachemo University, Hosaena, Ethiopia, **2** Department of Public Health, College of Health Science, Debre Berhan University, Debre Berhan, Ethiopia, **3** School of Public Health, Curtin University, Perth, Western Australia, Australia, **4** School of Medicine, College of Health Sciences and Medicine, Wolaita Sodo University, Wolaita Sodo, Ethiopia

* kehabtimershfrw@gmail.com

## Abstract

### Background

Short inter-pregnancy interval is an interval of <24 months between the dates of birth of the preceding child and the conception date of the current pregnancy. Despite its direct effects on the perinatal and maternal outcomes, there is a paucity of evidence on its prevalence and determinant factors, particularly in Ethiopia. Therefore, this study assessed the prevalence and associated factors of short inter-pregnancy interval among pregnant women in Debre Berhan town, Northern Ethiopia.

### Methods

A community based cross-sectional study was conducted among a randomly selected 496 pregnant women in Debre Berhan town from February 9 to March 9, 2020. The data were collected by using an interviewer-administered questionnaire and analyzed using STATA (14.2) statistical software. To identify the predictors of short inter-pregnancy interval, multivariable binary logistic regression was fitted and findings are presented using adjusted odds ratio (AOR) with 95% confidence interval (CI).

### Result

The overall prevalence of short inter-pregnancy interval (<24 months) among pregnant women was 205 (40.9%). Being over 30 years of age at first birth (AOR = 3.50; 95% CI: 2.12–6.01), non-use of modern contraceptive (AOR = 2.51; 95% CI: 1.23–3.71), duration of breastfeeding for less than 12 months (AOR = 2.62; 95% CI: 1.32–5.23), parity above four (AOR = 0.31; 95% CI: 0.05–0.81), and unintended pregnancy (AOR = 5.42; 95% CI: 3.34–9.22) were independently associated factors with short inter-pregnancy interval.

### Conclusion

Despite the public health interventions being tried in the country, the prevalence of short inter-pregnancy interval in this study is high. Therefore, it implies that increasing

**Data Availability Statement:** All relevant data are within the manuscript and its Supporting information files.

**Funding:** Debre Berhan University (www.dbu.edu.et) funded this study. The funder had no role in the study design, data collection and analysis, decision to publish, or preparation of the manuscript.

**Competing interests:** The authors have declared that no competing interests exist.

contraceptive use and encouraging optimal breastfeeding might help in the efforts made to avert the problem.

## Introduction

The inter-pregnancy interval is the time between the birth of the preceding child and the conception date of the current pregnancy. A short inter-pregnancy interval is when the interval between the delivery date of the preceding live birth and the conception date of the index birth is less than 24 months [1].

Historically, the World Health Organization (WHO) and other international authorities had recommended at least 2 to 3 years between successive pregnancies, and the United States Agency for International Development (USAID) had suggested an interval of 3 to 5 years [1]. Given the inconsistency, various countries and regional programs requested the WHO to further review the research and provide recommendations. As a result, the report of the 2005 WHO Technical Consultation and Scientific Review of Birth Spacing recommended waiting at least 2 years after a live birth and 6 months after miscarriage or induced termination before conception of another pregnancy [1].

A short inter-pregnancy interval (IPI) is associated with adverse maternal and infant health outcomes. It is known to hurt perinatal, neonatal, and child health outcomes, including preterm birth, low birth weight, perinatal death, still birth, intellectual disability, and developmental delay. Besides, it has also adverse maternal health outcomes such as nutritional depletion, anemia, cervical insufficiency, antepartum hemorrhage, premature rupture of membrane, and eclampsia [2–6].

In low-income countries, the prevalence of short inter-pregnancy interval ranges from 19.4% to 65.9% [4,7]. Even though there are studies in developed and some low-income countries, there is a paucity of evidence on the prevalence and predictors of inter-pregnancy interval in Ethiopia regardless of the country's culture encouraging women to have many children. On top of this, there are limitations of the currently available literature.

Most of the researches used birth interval, a proxy measure of time between two consecutive births, which could under or overestimate the time interval between the birth date of the preceding child and the conception date of the pregnancy. However, this study used the inter-pregnancy interval which measures correctly the time elapsed between the date of birth of the preceding child and the conception date of the current pregnancy. Therefore, the result of this study could present a more accurate picture of the problem and aid in the efforts being tried to reduce the short inter-pregnancy interval. The aim of this study was to assess the prevalence and associated factors of short inter-pregnancy interval among pregnant women in Debre Berhan town, Northern Ethiopia.

## Methods and materials

### Study setting, design and period

This community based cross-sectional study was conducted in Debre Berhan town, North Shewa zone, Northern Ethiopia from February 9 to March 9, 2020. We selected Debre Berhan town because it was the largest town in North Shewa zone where we can find the larger proportion of women compared to other towns of the zone and considering the available budget to conduct the study. The town is located 130 km from Addis Ababa, and it has nine kebeles. Kebele is the smallest administrative region in Ethiopia, approximately comprising 1,000

households. The total number of populations in the town is 57,787. The characteristics of the population in all the 9 kebeles is more or less similar. There are 795 pregnant women in these kebeles who are registered by health extension workers. In the town, there are 29 health institutions: two hospitals (one private and one referral governmental hospital), 3 health centers, 17 drug stores, and 7 private specialty clinics. In the health centers and hospitals of Ethiopia, midwives are the primary point of care for all pregnancies.

## Populations

Pregnant women who gave birth at least once (uniparous) and live in Debre Berhan town were the source population of this study. Women who had a miscarriage/abortion immediately before the current pregnancy were excluded from the study because they are more likely to get pregnant earlier as a result of the pregnancy loss, and they are recommended to have an interval of only six months.

## Sample size determination and sampling procedure

The sample size was determined using Epi info version 7 considering 28.5% prevalence of short inter-pregnancy interval from the study done in Bahir Dar Felegehiwot Hospital [8]. The final sample size was determined as 517 after considering a design effect of 1.5 and a non-response rate of 10%. Simple random sampling technique was used to select the kebeles and the participants. Among the nine kebeles, five kebeles were randomly selected. Then the sample size was proportionally allocated to those five kebeles. Family folder from the hands of Health Extension Workers (HEWs) of each kebele was used as a sampling frame to obtain a list of pregnant women in each kebele, and computer-generated random numbers were identified by the principal investigator and printed out to be given to the data collectors. Then, the data collectors visited the household of the study participants to conduct the interview.

## Data collection procedure and quality management

A structured interviewer administered questionnaire was prepared and implemented after reviewing relevant literature. The questionnaire was prepared in English then translated into the local language (Amharic) and finally translated back to English to check its consistency. It consists of socio-demographic, reproductive and health service related factors. It was checked and pre-tested in 5% of the study population outside the selected kebeles. After training was given for two days, the data were collected by five diploma midwives through a home-to-home visit. Revisits of two to three times were made for women who were not available at the time of the survey. The collected data were checked for completeness and consistency on each day of data collection. Supervision and monitoring were made every day by the assigned supervisors and the principal investigator.

## Measurement

Inter-pregnancy interval was defined as the time in completed months from the reported date of live birth of the previous child to the self-reported last normal menstrual period (LNMP). Most participants knew the date of birth of the previous child and last normal menstrual period of the current pregnancy. However, in case of the participants who didn't know the specific date of conception and/or the birth date of the previous child, the mid-date of the month was taken as the birth date of the previous child or the date of the conception for the current pregnancy. Therefore, the inter-pregnancy interval was calculated by subtracting the date of birth of the last child (previous child) from the date of conception of the current pregnancy

(IPI = date of conception (LMP)—date of birth of the previous child). So, short inter-pregnancy interval was defined as an interval less than 24 months.

## Data processing and analysis

The data were entered, cleaned and processed by Epi-data version 3.1 software and exported to STATA version 14.2 for analysis. (The analyzed data are provided with S1 File). Descriptive statistics such as frequencies, proportions and summary statistics were used to describe the study population with relevant variables. Association between the outcome and explanatory variables was assessed by using a binary logistic regression model. Variables with a p-value of $\leq 0.2$ in bivariable analysis were entered together into the model to conduct a multivariable analysis so as to control their effects of confounding. Statistical significance was considered at a level of significance of 5%, and adjusted odds ratio along with a 95% confidence interval was used to present the estimates of the strength of the associations. Hosmer-Lemeshow and variance inflation factor (VIF) was used to test the model fitness and multicollinearity respectively.

## Ethical consideration

Ethical clearance was obtained from the Ethical Review Committee of the Debre Berhan University. Then a permission letter from the Debre Berhan town health office and Debre Berhan town administration office was obtained. Moreover, the informed written consent was obtained from each respondent. Personal identifier such as name was not mentioned in the questionnaire.

## Results

### Socio-demographic characteristics of the study participants

A total of 496 pregnant women was included in the study yielding a response rate of 96%. The age of the participants ranged from 20 to 42 with the mean age (±SD) of 29.5 (±4.7) years. Of the study participants, around half (52.22%) were between the age of 25 to 29 years. One hundred seventy-three (34.88%) of the participants had attended college and above in their educational status, and 413 (83.26%) were orthodox Christians by religion (Table 1).

### Reproductive and health service related factors of the study participants

One hundred ninety-six (39.52%) of the participants did not use modern contraceptive before the current pregnancy, and 108 (21.77%) participants had an unintended pregnancy. Forty-eight (9.67%) of the study participants had no antenatal care (ANC) follow up by skilled attendants during the pregnancy of the index child. Similarly, forty-nine (9.87%) of the participants provided breastfeeding of their index child for only less than 12 months. Seventy-one (14.31%) participants had children of four and above, excluding the current pregnancy (Table 2).

### Prevalence of short inter-pregnancy interval

The prevalence of short inter-pregnancy interval (< 24 months) of this study was 205 (40.9%) with 95% CI: 36.6 to 45.4%. The median (IQR) inter-pregnancy interval of the study participants was 29 (18, 48) months. Of those who had a short inter-pregnancy interval, 24 (5%) had very short inter-pregnancy interval (<12 months). Besides, 210 (42%) and 86 (17.2%) of the participants had an inter-pregnancy interval of 24 to 59 months and more than 60 months, respectively.

**Table 1. Socio-demographic characteristics of the study participants, Debre Berhan town, Amahra region, Ethiopia, 2020.**

| Characteristics | Frequency (n = 496) | Percent (%) |
|---|---|---|
| **Current maternal age** | | |
| 20–24 years | 43 | 8.66 |
| 25–29 years | 259 | 52.22 |
| 30–34 years | 110 | 22.17 |
| 35–39 years | 61 | 12.29 |
| $\geq$ 40years | 23 | 4.63 |
| **Maternal age at first birth** | | |
| $\leq$ 30 years | 415 | 83.67 |
| >30 years | 81 | 16.33 |
| **Religion** | | |
| Orthodox | 413 | 83.26 |
| Muslim | 39 | 7.86 |
| Protestant | 44 | 8.87 |
| **Current maternal occupational status** | | |
| Self employed | 81 | 16.33 |
| Private employee | 76 | 15.32 |
| Government employee | 147 | 29.64 |
| Housewife | 183 | 36.89 |
| Student | 9 | 1.81 |
| **Husband's occupation** | | |
| Self employed | 155 | 31.25 |
| Private employee | 86 | 17.34 |
| Government employee | 236 | 47.58 |
| Other | 19 | 3.83 |
| **Household monthly income** | | |
| < 1000 ETB | 27 | 5.44 |
| 1000–2999 | 46 | 9.27 |
| 3000–4999 | 95 | 19.15 |
| > 5000 | 328 | 66.13 |

## Factors associated with short inter-pregnancy interval

Bivariable and multivariable logistic regression analyses were carried out to determine the association between the explanatory variables and short inter-pregnancy interval. Hence, based on the p-value ($< 0.2$) of the bivariable analysis, current maternal age, age at first birth, parity, unintended pregnancy, non-use of modern contraceptive before the current pregnancy, duration of breastfeeding, and survival status of the index child were selected as candidate variables to be included in the final model. However, the result of multivariable analysis confirmed that age at first birth, parity, unintended pregnancy, non-use of modern contraceptive before the current pregnancy, and duration of breastfeeding were independently associated with short inter-pregnancy interval. Multicollinearity was checked using a variance inflation factor and yielded a result of <10 for all variables in the final model (Table 3).

## Discussion

A community based cross-sectional study was conducted to assess the prevalence and associated factors of the short inter-pregnancy interval among pregnant women of the Debre Berhan

**Table 2. Reproductive and health service related factors of pregnant women in Debre Berhan town, Amahra region, Ethiopia, 2020.**

| Characteristics | Frequency (n = 496) | Percent (%) |
|---|---|---|
| **Use of contraceptive before the current pregnancy** | | |
| Yes | 300 | 60.48 |
| No | 196 | 39.52 |
| **ANC visit for the index child** | | |
| Yes | 448 | 90.32 |
| No | 48 | 9.67 |
| **Exclusive breastfeeding for the index child** | | |
| < 2 months | 87 | 17.54 |
| 2–3 months | 74 | 14.92 |
| 4–5 months | 69 | 13.91 |
| 6–7 months | 253 | 51.00 |
| Above 7 months | 13 | 2.62 |
| **Total duration of breastfeeding for the index child** | | |
| ≤12 months | 49 | 9.87 |
| 13–23 months | 136 | 27.42 |
| 24 and above months | 311 | 62.70 |
| **Parity** | | |
| <4 | 425 | 85.68 |
| ≥4 | 71 | 14.31 |
| **Pregnancy intention** | | |
| Intended | 388 | 78.22 |
| Unintended | 108 | 21.77 |
| **Survival status of the index child** | | |
| Alive | 480 | 96.77 |
| Dead | 16 | 3.22 |
| **Sex of the index child** | | |
| Female | 262 | 52.82 |
| Male | 234 | 47.18 |
| **Menstrual cycle pattern** | | |
| Regular | 283 | 57.05 |
| Irregular | 213 | 42.94 |
| **History of infertility** | | |
| Yes | 26 | 5.24 |
| No | 470 | 94.75 |
| **Mode of delivery of the index child** | | |
| Vaginal | 431 | 86.89 |
| Cesarean section | 65 | 13.10 |

town. Consequently, the overall prevalence of short inter-pregnancy interval (< 24 months) among pregnant women was 205 (40.9%). The factors independently associated with short inter-pregnancy interval were age at first birth, parity, unintended pregnancy, non-use of modern contraceptive before the current pregnancy, and duration of breastfeeding.

The prevalence in this study is higher than the studies in Bahidar, Felegehiwot Hospital [8] and the United States (US) [9] where about 28.5% and 35% of women had short inter-pregnancy interval respectively. This difference might be attributed to the cut off point for short inter-pregnancy interval. In this study, the cut off point for short inter-pregnancy interval

**Table 3. Bivariable and multivariable binary logistic regression analyses results of factors associated with short inter pregnancy interval among pregnant women in Debre Berhan town, Amahra region, Ethiopia, 2020.**

| Variable | Short inter-pregnancy interval (in months) | | COR (CI: 95%) | AOR (CI: 95%) | p-value |
|---|---|---|---|---|---|
| | Yes (n, %) | No (n, %) | | | |
| **Current age of respondent** | | | | | |
| 20–24 years | 24(55.81) | 19 (44.19) | 1.94(1.01–3.72)* | 1.84(0.836–4.04) | 0.1 |
| 25–29 years | 104(40.15) | 155(59.85) | 1 | 1 | - |
| 30–34 years | 47(42.73) | 63(57.27) | 1.15 (0.73–1.80) | 1.41(0.84–2.36) | 0.2 |
| 35–39 years | 21 (34.43) | 40(65.57) | 0.81 (0.45–1.44) | 1.06(0.53–2.11) | 0.8 |
| >= 40 years | 9(39.13) | 14(60.87) | 0.98(0.41–2.36) | 1.49(0.49–4.47) | 0.5 |
| **Age at first birth** | | | | | |
| <= 30 years | 151 (36.39) | 264(63.61) | 1 | 1 | - |
| >30 years | 54 (66.66) | 27 (33.33) | 3.61 (2.15–5.89) * | 3.50(2.12–6.01) | <0.001* |
| **Use of contraceptive before the current pregnancy** | | | | | |
| Yes | 115(36.98) | 196(63.02) | 1 | 1 | - |
| No | 90(48.65) | 95(51.35) | 2.05(1.42–2.96) * | 2.51(1.23–3.71) | 0.007* |
| **Duration of breastfeeding for the index child** | | | | | |
| ≤12 months | 29(59.18) | 20(40.82) | 2.46 (1.33–4.55) * | 2.62(1.32–5.23) | 0.006* |
| 13–23 months | 59(43.38) | 77(56.62) | 1.30 (0.86–1.96) | 1.21(0.74–1.95) | 0.4 |
| 24 and above months | 117(37.62) | 194(62.38) | 1 | 1 | - |
| **Parity** | | | | | |
| <4 | 164(38.58) | 261(61.41) | 1 | 1 | - |
| ≥4 | 41(57.74) | 30(42.25) | 0.17(0.08–0.35) * | 0.31(0.05–0.81) | <0.001* |
| **Pregnancy intention** | | | | | |
| Intended | 133 (34.27) | 255(65.72) | 1 | 1 | - |
| Unintended | 72 (66.66) | 36(33.33) | 3.91(2.48–6.14) * | 5.42(3.34–9.23) | <0.001* |
| **Survival status of the index child** | | | | | |
| Alive | 193(40.21) | 287(59.79) | 1 | 1 | - |
| Dead | 12 | 4 | 4.54(1.44–14.27) * | 2.97(0.80–11.03) | 0.1 |

* = statically significant at p-value of ≤ 0.05, COR = crude odds ratio, AOR = adjusted odds ratio.

was < 24 months. In comparison, the study conducted in the US and rural health center Manga Mandi, District Lahore defined short inter-pregnancy interval to be less than 18 months [9,10]. On the other hand, this finding is lower compared to the study conducted in Nigeria [7] and Selangor [11] where the prevalence of short inter-pregnancy interval is 65.9% and 48% respectively. This difference might be attributed to the sample population and socio-cultural practice.

In this study, the odds of experiencing short inter-pregnancy interval was 3.5 times higher among women who started child bearing above the age of 30 years compared to those who start at 30 years of age and lower. This finding is consistent with the study done in Bahirdar (Felegehiwot hospital) [8], and the US [9,12]. This might be due to the intention to use the remaining fertility age efficiently before the woman reaches the stage of menopause. In line with the evidence from two studies done in Nigeria [7,13], the finding of this study revealed that women who did not use modern contraceptive before the current pregnancy had 2.5 times higher odds of experiencing short inter-pregnancy interval as compared to those who used it. This can be explained by the potential of modern contraceptive to prevent and extend pregnancy.

This study also found out unintended pregnancy to be associated with short inter-pregnancy interval. The odds of experiencing short inter-pregnancy interval was 5.4 times higher among women with unintended current pregnancy compared to their counterparts. This finding is congruent with the study conducted in the US [9] and Selangor [11]. This might be due to a woman who plan to be pregnant may follow the recommendation for child spacing and therefore end up with optimal inter-pregnancy interval. "Since non-utilization and failure of contraceptive are among the major contributors of the unintended pregnancy, this might have contributed to the shortened inter-pregnancy interval" [14]. This study revealed that the odds of short inter-pregnancy interval was 2.6 times higher among women who breastfed their last child for less than 12 months compared to those who breastfed for 24 and above months. This finding is in line with the evidence from the study done in Nigeria [7]. It might be due to the fact that the duration of breastfeeding, including exclusive breastfeeding improves infant survival and lengthens the interval between pregnancies due to lactational amenorrhea (negative hormonal feedback). During breastfeeding, the receptors in the breast nipple will be stimulated, and this initiates a signal to the hypothalamus: a nerve center in the brain, which in turn signals the pituitary gland, thereby inhibits ovulation by reducing the release of gonadotrophic hormone needed for ovulation which results in post-partum amenorrhea [15].

The study also showed that parity was negatively associated with short inter-pregnancy interval. Women who had four and above children had 70% lower odds of experiencing short inter pregnancy interval compared to the counter groups. This finding is in line with the study done in Nigeria [13] and rural Bangladesh [16], but in contrast with the study done in Selangor [11]. These women may have achieved their desired family size and may feel less pressure or may be in a less hurry to get pregnant again.

## Limitation

The inter-pregnancy interval and breast-feeding duration were calculated based on women recall, which might result in recall bias. Being the data obtained through self-report of the women, the accuracy might not be at a level obtained objectively, even though respondents were critically informed about giving accurate information through assuring the confidentiality of their responses. The exclusion of women who experienced miscarriage/abortion immediately before the current pregnancy might have underestimated the prevalence of the short inter pregnancy interval.

## Conclusion

The World Health Organization (WHO) and the government of Ethiopia recommended that a woman should wait 24 months before attempting the next pregnancy after a live birth. Despite this recommendation, this study found out a higher proportion of women (40.9%) getting pregnant before the recommended period of time. Age at the first birth, parity, non-use of modern contraceptive, duration of breastfeeding, and unintended pregnancy were independently associated with short inter-pregnancy interval in the study. Therefore, it implies that increasing contraceptive use and encouraging optimal breastfeeding might help in the efforts made to avert the problem. Besides, further studies in the rural setup with higher sample size are needed to ascertain the prevalence and determinants of short inter-pregnancy interval.

## Supporting information

**S1 File. Dataset of the study.**
(XLSX)

**S2 File. Questionnaire of the study.**
(DOCX)

## Author Contributions

**Conceptualization:** Hana Mamo, Kehabtimer Shiferaw Kotiso.

**Data curation:** Hana Mamo, Kalayu Brhane, Kehabtimer Shiferaw Kotiso.

**Formal analysis:** Hana Mamo, Kalayu Brhane, Kehabtimer Shiferaw Kotiso.

**Funding acquisition:** Hana Mamo.

**Methodology:** Hana Mamo, Abinet Dagnaw, Nigussie Tadesse Sharew, Kalayu Brhane, Kehabtimer Shiferaw Kotiso.

**Project administration:** Hana Mamo.

**Software:** Hana Mamo, Kehabtimer Shiferaw Kotiso.

**Supervision:** Abinet Dagnaw, Nigussie Tadesse Sharew, Kehabtimer Shiferaw Kotiso.

**Validation:** Abinet Dagnaw, Nigussie Tadesse Sharew, Kalayu Brhane.

**Writing – original draft:** Hana Mamo, Kehabtimer Shiferaw Kotiso.

**Writing – review & editing:** Abinet Dagnaw, Nigussie Tadesse Sharew, Kalayu Brhane, Kehabtimer Shiferaw Kotiso.

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
