## [Decision Letter · Decision Letter 0]

13 Apr 2021

PONE-D-21-08455

Prevalence of short interpregnancy interval and its associated factors among pregnant women in Debre Berhan town, Ethiopia

PLOS ONE

Dear Dr. Kotiso,

Thank you for submitting your manuscript to PLOS ONE. After careful consideration, we feel that it has merit but does not fully meet PLOS ONE’s publication criteria as it currently stands. Therefore, we invite you to submit a revised version of the manuscript that addresses the points raised during the review process.

Please submit your revised manuscript by30 April. If you will need more time than this to complete your revisions, please reply to this message or contact the journal office at plosone@plos.org. Please include the following items when submitting your revised manuscript:

We look forward to receiving your revised manuscript.

Kind regards,

Claudia Marotta

Academic Editor

PLOS ONE

Journal Requirements:

Please include additional information regarding the survey or questionnaire used in the study and ensure that you have provided sufficient details that others could replicate the analyses. For instance, if you developed a questionnaire as part of this study and it is not under a copyright more restrictive than CC-BY, please include a copy, in both the original language and English, as Supporting Information. Moreover, please include more details on how the questionnaire was pre-tested, and whether it was validated.

Please upload a copy of Supporting Information File 1 which you refer to in your text on page 15.

Additional Editor Comments :

dear authors follow reviewer suggestions to improve your paper

Reviewers' comments:

Reviewer's Responses to Questions

**Comments to the Author**

1. Is the manuscript technically sound, and do the data support the conclusions?

Reviewer #1: Yes

Reviewer #2: Partly

2. Has the statistical analysis been performed appropriately and rigorously? 

Reviewer #1: Yes

Reviewer #2: I Don't Know

3. Have the authors made all data underlying the findings in their manuscript fully available?

Reviewer #1: Yes

Reviewer #2: Yes

4. Is the manuscript presented in an intelligible fashion and written in standard English?

Reviewer #1: Yes

Reviewer #2: No

5. Review Comments to the Author

Reviewer #1: PLOS One Review: Prevalence of short interpregnancy interval and its associated factors among pregnant women in Debre Berhan Town, Ethiopia

Dear PLOS One Editorial Board and authors of this manuscript,

I would first like to acknowledge and commend the work preformed by this group. The questions they asked are of particular importance, not only in Debre Berhan Town but globally as well, and paramount to the understanding of maternal health and development of children. Without first an understanding of the frequency of and factors that lead to a short interpregnancy interval it is impossible to then work on making changes, improve health and awareness for the society and future research projects. Additionally, I find it personally impressive that they were able to enroll and receive responses from nearly 500 people in a short, one month time frame – an indication in my mind that they are committed and focused on applying the data and making true change in their community not merely publishing another paper. Further, a 500 individual sample size is of course quite robust.

The goal of this paper was, as is clearly stated in their title, to determine how common a short interpregnancy interval is and what variables are correlated and associated to it specifically in Debre Berhan Town, Ethiopia. They preformed a cross sectional study using 5 midwives to sample pregnant mothers in 5 of 14 randomly selected kebeles (which my understanding act as areas/neighborhoods within Debre Berhan Town) and then pregnant mothers were sampled proportionately among those 5 kebeles at random. These mothers were then given a questionnaire administered by the midwives, each of whom sampled one kebele. For the purpose of pregnancy, abortions and miscarriages were not counted which I will discuss further in a moment. This study found that 40.9% of sampled women were pregnant again within this World Health Organization suggested period. Per their clearly written conclusion, “Age at the first birth, parity, non-use of modern contraceptive, duration of breast feeding, and unintended pregnancy were independently associated with short inter-pregnancy interval in the study. Therefore, it implies that increasing contraceptive use and encouraging optimal breast feeding might help in the efforts made to avert the problem.”

This manuscript is written clearly and such that non-specialists in the field can certainly understand, but some clarification with regard to customs and structure within Debre Berhan Town, Ethiopia would be beneficial as is discussed further bellow.

Before moving forward and addressing the areas left for improvement and questions that should be asked or considered in this paper, I would first like to make two points which may explain some of the points I am about to make:

1) I acknowledge that the patient data was collected from February 9th to March 9th of 2020, ending just a few days prior to the WHO declaration that there was a global pandemic caused by sars-cov-2 (COVID-19). It is conceivable that a larger study was originally planned.

2) I am not entirely familiar with what life is like in Debre Berhan Town, Ethiopia – as I imagine is the case for most people who will read this publication. As such, I may be missing some information that is needed to fully understand the authors position or the nature of pregnancy care and labor. While I was able to find some of this information online, such as population size and what definition of a Kebele, more of this information should be stated clearly in the manuscript.

Here are my points for improvement and/or clarification:

1) Why only Debre Berhan Town, Ethiopia?

a. I can make the assumption that this is because this is where the authors work and where their funding is from, however, it should be stated clearly.

b. Is this town particularly at risk for having a low interpregnancy duration?

2) With regard to Kebeles and addressing sample representation of Debre Berhan Town, Ethiopia:

a. Define a Kebele better early in the paper

b. Determine whether or not the 5 kebeles – while randomly selected – are indicative of the general population.

i. Ie. Were they the wealthiest or poorest kebeles? Do all kebeles have similar structural make up? Do all kebeles have access to the same healthcare? There are some socioeconomic techniques that can be used to preform these statistics if not readily available.

3) Describe access to healthcare in Debre Berhan Town and how women normally give birth, particularly in relation to the fact that in this study only midwives were used to gather all of the data (important for generalization ability and potential sources of bias):

a. What percentage of women give birth with a midwife as opposed to physician, family alone, c-section etc?

b. Do midwives take care of all pregnancies?

i. Ie if a mother is particularly ill or there is a complication with the pregnancy, does a midwife provide the care? If not, then there is selection bias against this group.

c. Paper should either be rewritten to reflect it is looking at this subset of pregnancies/births if this is the case or address other forms of birth as well.

4) Perhaps some co-variables could be looked at together, not simply alone.

5) “Family Planning” and interventions are mentioned in introduction. This raises the following points for this manuscript:

a. It is mentioned that the interpregnancy period is short, despite these interventions. Was there data from before that the new data presented in this paper can be presented against?

b. Family planning does not seem to be discussed further in the manuscript but should be considered as a variable, unless everyone has this training – in which case that needs to be stated.

6) Abortions and miscarriages were not counted as pregnancies, but equally was also not considered as a variable which seems short sighted as it can be incredibly impactful on a mother and family.

a. Excluding these cases can lead to under reporting the data and leading to the data, and thus study, being less generalizable.

b. Why were they excluded?

i. Would the following situation be considered as a pregnancy interval: Birth 1, miscarriage, birth 2? Would the mother be excluded from the study? Or would the miscarriage merely not be considered?

c. What is the abortion rate and miscarriage rate in Debre Berhan Town, Ethiopia?

d. I do acknowledge that infant mortality rates and maternal infertility history are considered and may reflect some of these concerns. If so, please discuss and address this fact more specifically.

To reiterate it is my belief that the authors conducted important research with an impressive work ethic and sample size, while focusing on what I ascertain to be important and actionable variables for their situation, which can likely be generalized to other areas of a similar nature.

There are no ethical concerns or statistical interpretation concerns.

It is my formal recommendation that the Editors accept this paper after minor revisions are completed to clarify the aforementioned points and I encourage the authors to continue this line of inquiry for the betterment of the community they are serving but also the global research community that is also focusing on the effects and associated factors of short interpregnancy time. There is potential for further, more in-depth research into the well-being of the mother and children from this study, as the impact of a short interpregnancy period is not studied beyond the introduction. Additionally, a sociological and psychological approach to understanding the influences behind a short interpregnancy interval would certainly be interesting and add another level of depth to this paper. However, this would change the nature of this manuscript significantly and is not needed to finalize a publishable product.

I wish the authors the best in their efforts to help their community and contribute meaningfully to the international research community. It has been a privilege to review their work, and I am appreciative of this opportunity.

Kindly,

The Reviewer

Reviewer #2: ABSTRACT: Under background, the definition of interpregnancy interval (1st sentence) should be recasted for better understanding. Grammatical errors that need to be corrected.

INTRODUCTION: Lines 5-7,page 3,paragraph 2 should be referenced. Page 3, paragraph 3,lines 14-18 should be recasted.

METHODS: Why was Debre Berham chosen as the study location? The authors should explain what kebeles are for international readers. There should be a more detailed explanation as to how the pregnant women were selected, where this was done, by whom and when? When were the questionnaires administered and where was this done?

DISCUSSION: Lines 8,page 11,US should be written in full 1st. Also RHC (lines 11,page 8.) In page 12,lines 8 and line 10 when referencing statements, the authors mix cities and countries in different references.eg Port-Harcourt (reference 13) is a town in Nigeria but reference 7 is Nigeria. Also US (reference 9) then Michigan reference 12. The referencing of statements should be uniform for better understanding. Page 12,line 13-14 should be recasted. The literature review should be more robust.

CONCLUSION: The statement in the last line in the conclusion in page 13, including the non-use of contraceptive as part of the recommendations reduce the incidence of short interpregnancy intervals seems to be at variance with the information in the last paragraph of the discussion before the limitations.

REFERENCES>NO 6- BMJ should be written in full

6. PLOS authors have the option to publish the peer review history of their article (what does this mean?). If published, this will include your full peer review and any attached files.

Reviewer #1: No

Reviewer #2: No

---

## [Author Response · Author response to Decision Letter 0]

7 May 2021

Dear reviewers and editor, 

Thanks for your review and important issues you raised in the paper. Hereunder, kindly get the responses to the points raised.

Response to editor

1. I have uploaded the questionnaire (both English version and Amharic version) as Supporting Information.

2. A copy of supporting information file 1 (the dataset of the study) was also uploaded in the system.

Responses to Reviewer #1

1. We selected Debre Berhan town because it was the largest town in North Shewa zone where we can find the larger proportion of women compared to other towns of the zone and considering the available budget to conduct the study. (stated in the manuscript)

b) Generally, the culture of Amhara region where Debre Berhan town is located, encourages giving birth of too many children

2. Kebele was defined as “Kebele is the smallest administrative region in Ethiopia approximately comprising 1,000 households.” in the manuscript. The characteristics of the population in all the 9 kebeles is more or less similar. (stated in the manuscript)

3. Relevant information in the study area including access to health care are included in the revised manuscript such as “In the town, there are 29 health institutions: two hospitals (one private and one referral governmental hospital), 3 health centers, 17 drug stores, and 7 private specialty clinics.” And in Ethiopia, the midwives give basic care to all pregnancies and refer those women with serious complications needing urgent referral to hospitals. To eliminate selection bias, the participants were selected randomly from the family folder, and the printed hardcopies were given to the data collectors to access the women and make interview. Besides, being a community-based study, the sampling will not depend on the level of care that the midwives will give.

4. We tried to check for possible interaction of variables; however, we haven’t found significant and/or plausible result.

5. The issue of family planning and interventions were also addressed in the revised manuscript. Actually, it was mentioned not in the background section but in the abstract subsection conclusion and now it’s fixed.

6. Women who had miscarriage/abortion immediately before the current pregnancy were excluded from the study because they are more likely to get pregnant earlier as a result of the pregnancy loss, and they are recommended to have an interval of only six months. Because of the difference in the recommended interval between the pregnancies for those who experienced abortion immediately before the current pregnancy and who didn’t, we excluded them not to overestimate the short interpregnancy interval in a biased manner.

However, those women who were recorded as pregnant in the family folder but had abortion at the study period were not excluded. Despite these, the possibility of bias associated with the exclusion was discussed in the limitation of the manuscript. 

Besides, we could not find a true figure of abortion rate and miscarriage rate in Debre Berhan Town as per our search. 

Responses to Reviewer #2

ABSTRACT: Under background, the definition of interpregnancy interval (1st sentence) was recast and corrected in the revised manuscript.

INTRODUCTION: Lines 5-7,page 3,paragraph 2 was referenced. Page 3, paragraph 3, lines 14-18 was recast.

METHODS: Concerning “Why was Debre Berham chosen as the study location?” And definition of kebeles, kindly see the responses to reviewer 1 under #1 and #2, respectively. 

A more detailed explanation as to how the pregnant women were selected was added in the revised manuscript. 

DISCUSSION: Lines 8,page 11,US was written in full 1st. Also RHC (lines 11,page 8.) In page 12, line 8 and line 10 the referencing issue, the authors mix cities and countries in different references was fixed, and the cities were replaced by the respective countries to make it consistent and for better understanding. 

Page 12, line 13-14 was recast. 

CONCLUSION: “The statement in the last line in the conclusion in page 13, including the non-use of contraceptive as part of the recommendations reduce the incidence of short interpregnancy intervals seems to be at variance with the information in the last paragraph of the discussion before the limitations.” Was addressed in the revised manuscript.

REFERENCES>NO 6- BMJ should be written in full. It is the journal’s actual name, and it’s how they suggest to cite their journal

With kind regards,

The authors

---

## [Decision Letter · Decision Letter 1]

22 Jun 2021

PONE-D-21-08455R1

Prevalence of short interpregnancy interval and its associated factors among pregnant women in Debre Berhan town, Ethiopia

PLOS ONE

Dear Dr. Kotiso,

Thank you for submitting your manuscript to PLOS ONE. After careful consideration, we feel that it has merit but does not fully meet PLOS ONE’s publication criteria as it currently stands. Therefore, we invite you to submit a revised version of the manuscript that addresses the points raised during the review process.

This manuscript has been improved by the various revisions made. Both reviewers have suggested further (minor) revisions. I think that they would improve the manuscript once again, so please attend to each one of the suggestions to make the manuscript as good as it can possibly be. I will not add to the various suggested revisions, which should not be too onerous for the authors to deal with. I look forward to seeing the next version of the manuscript.

We look forward to receiving your revised manuscript.

Kind regards,

Clive J Petry, PhD

Academic Editor

PLOS ONE

Journal Requirements:

Reviewers' comments:

Reviewer's Responses to Questions

**Comments to the Author**

1. If the authors have adequately addressed your comments raised in a previous round of review and you feel that this manuscript is now acceptable for publication, you may indicate that here to bypass the “Comments to the Author” section, enter your conflict of interest statement in the “Confidential to Editor” section, and submit your "Accept" recommendation.

Reviewer #1: (No Response)

Reviewer #2: (No Response)

2. Is the manuscript technically sound, and do the data support the conclusions?

Reviewer #1: Yes

Reviewer #2: Yes

3. Has the statistical analysis been performed appropriately and rigorously? 

Reviewer #1: Yes

Reviewer #2: I Don't Know

4. Have the authors made all data underlying the findings in their manuscript fully available?

Reviewer #1: Yes

Reviewer #2: Yes

5. Is the manuscript presented in an intelligible fashion and written in standard English?

Reviewer #1: Yes

Reviewer #2: No

6. Review Comments to the Author

Reviewer #1: Dear Authors,

Thank you for addressing my comments thoroughly and making appropriate adjustments to the manuscript. Please see the last few areas of improvement after which, I believe this manuscript will be ready for publication.

Introduction:

In the sentence "Therefore, the result of this study could present a true picture of the problem and aid in the efforts being tried to reduce the short inter-pregnancy interval." I suggest replacing the word "true" with "more accurate"

Study setting/Background:

Mention the culture often leads to women having too many children

Mention midwives are primary point of care for all pregnancies

Response 3: sampling may not change but it is important for the generalizability and validity of the paper.

Response 5 - thank you, I apologize for missing it.

Response 6 -thank you for the thorough further explanation.

I look forward to reading the final draft. Again, I note the importance of the research the authors have undertaken.

Kindly,

The Reviewer

Reviewer #2: GENERAL: Few grammatical errors and wrong tenses

METHODS: Under population (page 4), the 1st sentence should better read- Pregnant women who give birth a least once----.

DICUSSION: In page 13,line 3-4; This study is in line with a study done in Nigeria (13, 16) and Bangladesh (16). Reference number 16 is quoted for a study done in both Nigeria and Bangladesh.

REFERENCES: Number 6-bmj should be in capital letters. The paging in reference number 7 is not complete

7. PLOS authors have the option to publish the peer review history of their article (what does this mean?). If published, this will include your full peer review and any attached files.

Reviewer #1: No

Reviewer #2: No

---

## [Author Response · Author response to Decision Letter 1]

16 Jul 2021

Dear reviewers and editor, 

Thank you for your review and important issues you raised in the manuscript entitled “Prevalence of short interpregnancy interval and its associated factors among pregnant women in Debre Berhan town, Ethiopia.” 

We have read the reviewer's comments carefully and hope that the revised version now submitted will be regarded as having enhanced the previous version. The authors very welcome the reviewer's comments and suggestions. These contributions have appreciably improved the final paper quality. Hereunder, kindly get the responses to the points raised under each point.

Response to reviewers

Reviewers' comments:

Reviewer's Responses to Questions

Comments to the Author

1. If the authors have adequately addressed your comments raised in a previous round of review and you feel that this manuscript is now acceptable for publication, you may indicate that here to bypass the “Comments to the Author” section, enter your conflict of interest statement in the “Confidential to Editor” section, and submit your "Accept" recommendation.

Reviewer #1: (No Response)

Reviewer #2: (No Response)

2. Is the manuscript technically sound, and do the data support the conclusions?

Reviewer #1: Yes

Reviewer #2: Yes

3. Has the statistical analysis been performed appropriately and rigorously?

Reviewer #1: Yes

Reviewer #2: I Don't Know

4. Have the authors made all data underlying the findings in their manuscript fully available?

Reviewer #1: Yes

Reviewer #2: Yes

5. Is the manuscript presented in an intelligible fashion and written in standard English?

Reviewer #1: Yes

Reviewer #2: No

6. Review Comments to the Author

Reviewer #1: 

Introduction:

In the sentence "Therefore, the result of this study could present a true picture of the problem and aid in the efforts being tried to reduce the short inter-pregnancy interval." I suggest replacing the word "true" with "more accurate"

Comment: true was replaced by more accurate based on the reviewer’s comment

Study setting/Background:

Mention the culture often leads to women having too many children

Mention midwives are primary point of care for all pregnancies

Comment: amendments were made based on the reviewer’s comment.

Response 3: sampling may not change but it is important for the generalizability and validity of the paper.

Comment: thank you for the important notice.

Response 5 - thank you, I apologize for missing it.

Response 6 -thank you for the thorough further explanation.

Reviewer #2: 

GENERAL: Few grammatical errors and wrong tenses

Comment: Some grammatical errors were fixed based on the reviewer’s comment.

METHODS: Under population (page 4), the 1st sentence should better read- Pregnant women who give birth a least once----.

Comment: Corrections were made based on the reviewer’s comment.

DICUSSION: In page 13,line 3-4; This study is in line with a study done in Nigeria (13, 16) and Bangladesh (16). Reference number 16 is quoted for a study done in both Nigeria and Bangladesh.

Comment: reference number 16 is removed from a study done in Nigeria.

REFERENCES: Number 6-bmj should be in capital letters. The paging in reference number 7 is not complete

Comment: bmj was capitalized, and the paging in reference 7 was made complete.

Thank you for your consideration. We look forward to hearing from you.

Sincerely,

The authors

---

## [Editor Report · Decision Letter 2]

21 Jul 2021

Prevalence of short interpregnancy interval and its associated factors among pregnant women in Debre Berhan town, Ethiopia

PONE-D-21-08455R2

Dear Dr. Kotiso,

We’re pleased to inform you that your manuscript has been judged scientifically suitable for publication and will be formally accepted for publication once it meets all outstanding technical requirements.

Kind regards,

Clive J Petry, PhD

Academic Editor

PLOS ONE
---

## [Editor Report · Acceptance letter]

23 Jul 2021

PONE-D-21-08455R2 

Prevalence of short interpregnancy interval and its associated factors among pregnant women in Debre Berhan town, Ethiopia 

Dear Dr. Kotiso:

I'm pleased to inform you that your manuscript has been deemed suitable for publication in PLOS ONE. Congratulations! Your manuscript is now with our production department. 

Kind regards, 

on behalf of

Dr. Clive J Petry 

Academic Editor

PLOS ONE